# Increased Rate of Yeast Cultivation from Packaged Beer with Environmentally Relevant Anaerobic Handling

Kira Pai,[a] Ginger Stout,[a] Theresa Zimmer,[a] Clayton Jacobs,[a] Helene Ver Eecke[a]

[a]Metropolitan State University of Denver, Biology, Denver, Colorado, USA

**ABSTRACT** Beer production necessitates oxygen exclusion for the proper packaging and aging of the beer. Standard operating procedures, including those for quality testing, involve culturing microbes from packaged beer exposed to atmospheric oxygen, despite the generalized fact that packaged beer is an anaerobic environment. Our research goal was to apply an environmentally relevant culturing approach to improve yeast cultivation from bottled beer by attempting to ameliorate transplant shock. This is applicable to uniquely scrutinous quality assurance/control objectives and/or to grand cultivation goals, such as ancient beer samples. Although yeasts have the genetic capacity of oxygen protection, their epigenetic/biochemical states within anaerobic packaging may not adequately protect all cells from reactive oxygen species (ROS) at the moment of opening. Soon after opening, beer yeasts were found to be catalase negative, indicating deficient protection from at least one ROS. The general reduction/inhibition of growth was observed when the beer yeast was exposed to ROS in media, and atmospheric bottle opening was found to expose beer yeast to significantly increased levels of ROS. Our primary finding is that different oxygen handling methodologies (aerobic/microaerophilic/anaerobic) significantly impact the viable *Saccharomyces* yeast recovery rates of Bamberger's Mahr's Bräu Unfiltered Lager. Immediate anaerobic handling improved cultivation success rates, with significantly higher colony forming units (CFU)/mL being cultured, and reduced the volume of beer required to recover viable yeast. Aerobic standard operating procedures have mainly been developed to harvest yeast on large volumetric samples and/or samples with high viable cell numbers, but these procedures may be suboptimal and may underrepresent potential viable cell numbers.

**IMPORTANCE** Procedures of beer production and packaging exclude oxygen to create a shelf-stable anaerobic environment, within which any viable organisms are stored. However, standard methodologies to cultivate microbes from such environments generally include opening in an oxygenated atmosphere. This study applies environmentally relevant culturing methods and compares the yeast recovery rates of beers handled in various oxygen conditions. When beer bottles were opened in anoxic conditions, higher colony counts were obtained, so a smaller volume of beer was required to recover viable cells. The yeast in beer, stored anaerobically, may not be biochemically prepared to fully protect cells from oxygen at the moment of opening. Negative catalase activity showed beer yeasts' vulnerabilities to reactive oxygen. Atmospheric opening may reduce viability, causing the underreporting of viable cells. Anaerobic opening could increase the odds of successfully detecting/cultivating viable cell(s) that are present, which is pertinent to uniquely stringent quality screens and ambitious culturing attempts from rare samples.

**KEYWORDS** beer, cultivation optimization, historical brews, quality assurance, quantitative culturing, yeasts

Address correspondence to Helene Ver Eecke, vereecke@msudenver.edu.

The authors declare no conflict of interest.

*[This article was published on 31 October 2022 with an error in reference 8. The reference was corrected in the current version, posted on 4 November 2022.]*

Beer has been produced and generally enjoyed for over 5,000 years (1) by utilizing four primary ingredients: water, yeast, hops, and fermentable carbohydrates from cereal grains, usually malted barley. Globally, beer is currently the most widely consumed alcoholic beverage

(2, 3), the third most consumed drink overall (3, 4), and had a reported trade value of 31 billion USD in 2021 (5). To establish market share, competitive beer producers are expected to consistently produce hyperdifferentiated and high-quality beers (2, 6). An improved protocol for cellular recovery from beer would aid in producing such competitive beers by enhancing the ability to culture novel applicable microbes and/or potentially contaminant microbes.

Fermented liquids are sealed within their ecological microcosms, and niche transplantation can hinder cultivation. The general approach of this endeavor is to ameliorate shock to increase microbial survivability and cultivation from beer. Ecological theory inversely relates the number of species and their functional redundancies to ecological stability (7). A mono-cultured/single-species beer would probably be significantly impacted by environmental shifts, as it would be unlikely to be resilient due to its minimal functional redundancy. The factors of the beer environment, in general, have been well-described previously (8): alcoholic ($\sim$50 g/L), moderately acidic ($\sim$pH 4), moderate/low nutrient availability ($\sim$150 mg/L free amino nitrogen, >1.01 specific gravity), decreased cellular constituents/compounds ($\sim$50 mg/L sterol, $\sim$20% dry weight glycogen, $\sim$70 ppm pyruvate), and an anaerobic state ($\sim$0 ppb $O_2$). Breweries have recognized oxygen exclusion as a top priority for the proper packaging and aging of beer to promote stability (1, 9–11). Although packaged beer is an anaerobic environment, the atmospheric handling of beer is standardly performed during its culturing, including the standard procedures for quality assurance/control.

Atmospheric exposure may be shocking to yeast within sealed anaerobic vessels because reactive oxygen species (ROS) are known to damage DNA, proteins, lipids, and other macromolecules (12, 13). *Saccharomyces cerevisiae* (beer yeast) does have the genetic capacity to protect itself from the toxic effects of oxygen in various ways: CuZn-superoxide-dismutase (SOD-1), MnSOD (SOD-2), glutathione, ascorbate, catalases, peroxidases, and metallothionein (14). The expression of these genes is strictly regulated, as determined by the precise conditions of the cell. Oxygen concentration has several cascading effects on *S. cerevisiae*. For example, a decrease of oxygen leads to the absence of heme, causing the positive and negative regulation of the expression of more than 25 genes in ways that downregulate/eliminate oxygen protection genes (15). Although the presence of oxygen risks cellular damage caused by ROS, oxygen is required for certain *S. cerevisiae* cellular processes, such as electron transport, proline uptake, and the biosynthesis of sterols, unsaturated lipids, and fatty acids (16, 17). Facultative organisms, such as *Saccharomyces*, must regulate their genetic capacities judiciously to maintain their viability and energetics within a particular environment.

Our research goal was to develop a methodology with an increased success of recovering and cultivating yeast from packaged beer, based on environmental relevancy. The fundamentals of this quantitative cultivation method are to aseptically open packaged beer, concentrate any cells, put the potentially viable yeast mass onto a supportive medium, incubate to promote cell growth, and enumerate colonies. Various permutations of oxygen handling methods (aerobic/microaerophilic/anaerobic) were tested in this way to evaluate the optimal method that yields the highest number of colonies in culture per unit volume of beer. Oxygen was considered to be potentially threatening to cell viability because, although yeast is a facultative organism with the genetic capacity of oxygen protection, its epigenetic/biochemical state within the anaerobic bottle may not optimally protect it from all ROS. Our general hypothesis is that protecting packaged beer yeast from oxygen during key protocol steps of culturing is environmentally relevant and will increase successful cultivation. In addition to this primary quantitative culturing, experiments to quantify oxidative stress (ROS detection, $O_2$ measurements, catalase assays, and ROS spot-tests) were carried out to gain an ecological understanding of bottled beer and the yeast within, as well as for suggestive insight into ecological shifts that could impact cultivation success rates.

## RESULTS

**Yeast recovery from Bamberger's Mahr's Bräu Unfiltered Lager (BMB) with various oxygen exposures.** Throughout these cultivation experimentations, experimental manipulations were not found to introduce contaminants. No growth was apparent on plates inoculated with concentrated, filter-sterilized, beer. Plates inoculated with concentrated BMB beer

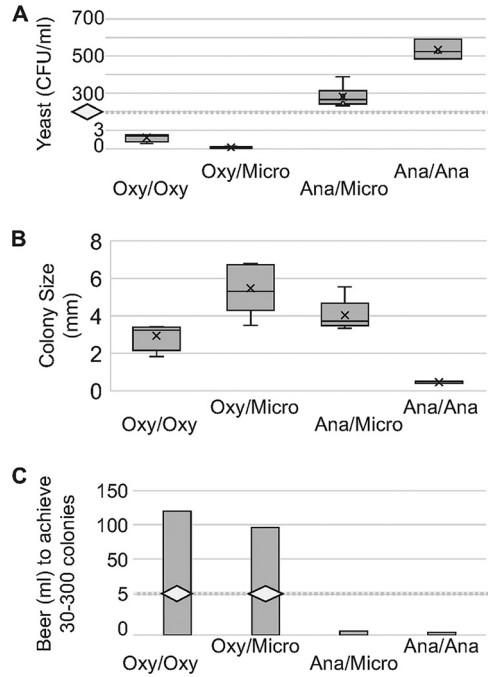

**FIG 1** Yeast recovery results with various oxygen handling methods during the opening of Bamberger's Mahr's Bräu Unfiltered Lager (BMB) bottles and incubation of cell concentrates on BTJ agar plates. Each condition is stated as opened/incubated oxygen handling method with 'Oxy' representing the aerobic state (~18% $O_2$), 'Micro' representing the microaerophilic state (~6% $O_2$), and 'Ana' representing the anaerobic state (~0% $O_2$). (A) Yeast recovery in CFU/mL in different opening/incubating atmospheres. More yeast colonies grew when the bottles were opened and when the cells were grown in lower oxygen conditions: oxy/oxy, 1.2 CFU/mL; oxy/micro, 0.3 CFU/mL; ana/micro, 280.3 CFU/mL; and ana/ana, 533.3 CFU/mL. (B) Average colony sizes of yeast grown in different oxygen conditions. Anaerobically grown yeast had smaller colonies: oxy/oxy, 2.9 mm; oxy/micro, 5.4 mm; ana/micro, 4.0 mm; and ana/ana, 0.5 mm. (C) Total volume of beer needed to achieve a countable number of colonies per plate. Significantly less beer is needed when the bottles are opened and when the cells are grown in low oxygen conditions: oxy/oxy, 120 mL; oxy/micro, 80 to 120 mL; ana/micro, 0.33 to 0.66 mL; and ana/ana, 0.25 to 0.33 mL.

exhibited only one colony type, which microscopic evaluation confirmed to be yeast. Bacterial cells were not cultured from BMB beer on Brewer's Tomato Juice (BTJ) agar.

Equal volumes of anaerobically opened beer produced confluent growth on solid medium, while there was no growth apparent in aerobically handled beer. To produce countable plates (30 to 300 colonies), the volumes of beer being processed needed to be dramatically altered for the different conditions of beer handling (Fig. 1; Fig. S1). Countable plates were achieved for all conditions, average colony counts were quantified, and average colony forming units (CFU)/mL were calculated based on the beer volume utilized (Fig. 1). The anaerobically incubated plates had small colonies (0.5 mm [±0.2]) compared to the other incubation conditions (2.9 to 5.4 mm). These anaerobically incubated colonies were distinct and represented the highest average CFU/mL, calculated to be 533 (±53.6) (Fig. 1; Fig. S1). Comparing the microaerophilically coincubated plates that were from beer samples opened aerobically versus anaerobically, there were clearly significant (*t* test, *P* value < 0.001) differences in CFU/mL with averages of 0.3 (±0.1) and 280.3 (±57.5), respectively (Fig. 1). This represents over a 900-fold decrease of cultivation when instantly exposed to oxygen.

All cultures were found to remain viable when transferred to conditions with higher oxygen concentrations. Microaerophilically incubated plates and colonies were found to transfer to aerobic conditions, and anaerobically incubated plates and colonies were found to transfer to microaerophilic and then to aerobic conditions. The colony sizes of the anaerobically incubated plates quickly increased from their relatively small average sizes of 0.5 mm to a comparable average size of 4.0 mm after only 1 day at microaerophilic conditions (Fig. S1).

**Relevant conditions within Bottles of Bamberger's Mahr's Bräu Unfiltered Lager (BMB) – Microbe(s), oxygen, and reactive oxygen species present.** Concentrated BMB beer was inoculated into BTJ broth to enrich the viable cells within. After 4 days, the turbid

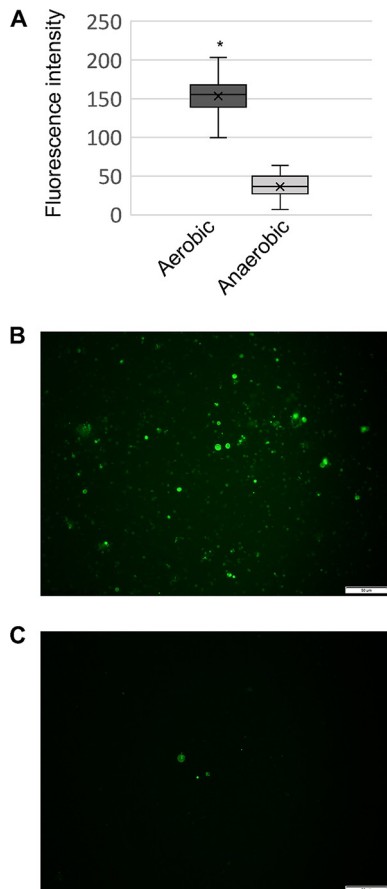

**FIG 2** Fluorescence detected with 2′7′-dichlorofluorescein diacetate (DCF) to visualize reactive oxygen species. Comparative quantification of cells' fluorescence (A) and representative microscopic images of Bamberger's Mahr's Bräu Unfiltered Lager yeast cells handled aerobically (B) and anaerobically (C).

enrichment was plated onto BTJ agar, and only one colony morphology was apparent. All colonies were observed with microscopy and confirmed to be yeast. BTJ media were prepared without antibiotics and have the potential to enrich for bacteria, but no bacteria were detected. DNA was successfully extracted from the primary BTJ broth enrichment, and fungal and yeast taxonomic genes were polymerase chain reaction (PCR) amplified. PCR was not able to amplify the bacterial 16SrDNA gene with the standard primers 27F and 1492R. Sanger methodology was able to sequence the PCR amplicons of the fungal taxonomy gene segment (ITS1 and ITS4 primers) and the yeast taxonomy gene segment (NL1 and NL4 primers), each of which indicated a single amplicon. The fungal taxonomy gene was found to be of the genus *Saccharomyces*, with the most closely related species being *S. cerevisiae* (beer yeast) (Fig. S2). This culture-based and molecular-based assessment of BMB beer suggests a monoculture of *Saccharomyces* yeast within the BMB beer.

An Anton Paar Packaged Beer Analyzer measured the BMB beer bottles to have standard beer attributes, including no detectable oxygen at the ppb level (Table S1). These measurements show that these bottles conform to industry packaging standards and exemplify that packaged beer products are generally anaerobic.

Utilizing the dye 2′7′-dichlorofluorescein diacetate (DCF) and fluorescence microscopy, as done previously (18), the relative amount of ROS within the BMB beer yeast was compared between beer opened aerobically and anaerobically. Utilizing uniform exposure settings, there were far fewer visible cells stained with DCF in the anaerobic sample, and the fluorescence of these cells was diminished in comparison to that of the cells in the aerobic sample (Fig. 2). Individual cells' mean gray values within the green channel of ImageJ were measured, and they displayed significant (*t* test, *P* value $< 0.001$) differences between yeasts

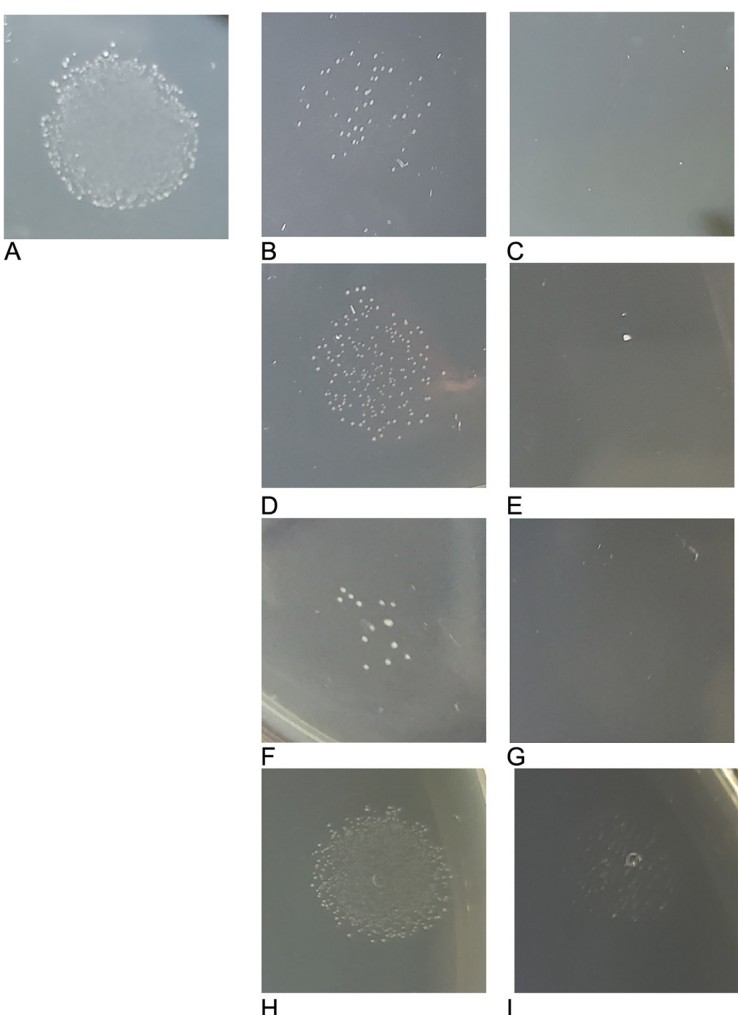

**FIG 3** Representational images of ROS spot-tests of yeast growth on standard YEPD medium (A), 3.75 mM H₂O₂ (B), 5.0 mM H₂O₂ (C), 1.25 mM tBHP (D), 2.5 mM tBHP (E), 1.0 mM diamide (F), 2.5 mM diamide (G), 225 mM menadione (H), and 300 mM menadione (I). Each image is of a standardized 8 μL aliquot of an anaerobically handled Bamberger's Mahr's Bräu Unfiltered Lager yeast concentrate. Each presented image is a standardized 18 mm of the plate.

treated aerobically 153 ($\pm$27) and yeasts treated anaerobically 36.0 ($\pm$16), (average [$\pm$ standard deviation]) (Fig. 2). These data support the idea that there are minimal ROS within sealed anaerobic packaged beer products and that exposure to aerobic conditions may quickly expose cells to increased ROS levels.

**Impact of reactive oxygen species on Bamberger's Mahr's Bräu Unfiltered Lager (BMB) yeast growth.** Conventional spot-tests were performed by depositing a uniform volume of inoculum and observing the diameter and opacity (integrated density) of the growth spot on agar after incubation. Variable amounts of chemical reagents were incorporated into the media to examine the bottled yeasts' physiologically relevant sensitivity to ROS. 8 μL of concentrated, anaerobically handled, BMB beer spotted on standard YEPD plates yielded dense growth spots with average diameters of 12.0 mm ($\pm$0.25) and average integrated densities of 20,892 ($\pm$1,797). The additional ROS had remarkable impacts on the growth of yeast compared to the standard control medium (Fig. 3). Strong growth inhibition, in terms of growth spot density, was recorded on spot-tests with either 3.75 mM hydrogen peroxide (H₂O₂), 1.25 mM tert-butyl hydroperoxide (tBHP), 1.5 mM diamide, or 225 mM menadione incorporated into YEPD plates. The average integrated densities for these 8 μL growth spots decreased from the control's 20,892 to 1,033 for the 3.75 mM H₂O₂, 2,734 for the 1.25 mM tBHP, 339 for the 1.5 mM diamide, and 3,766 for the 225 mM menadione. Growth was observed to be entirely inhibited with either 5 mM H₂O₂, 2.5 tBHP, or 25 mM diamide.

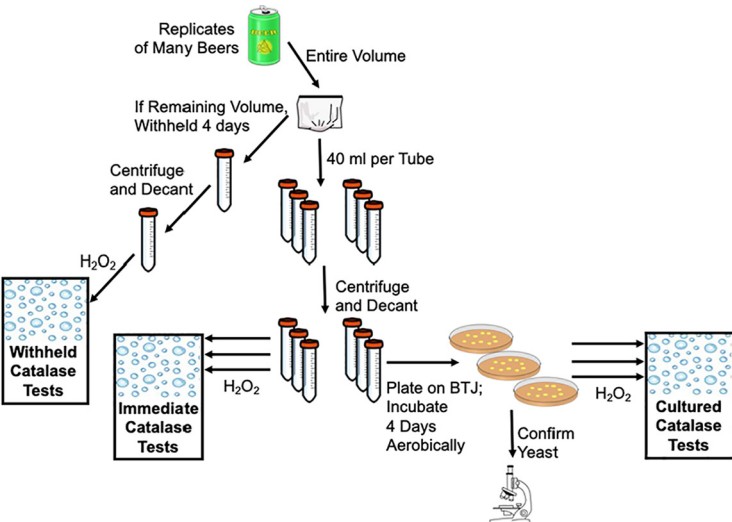

**FIG 4** Visual overview of catalase testing workflow. Various beer samples were tested immediately and after culturing aerobically for 4 days. Select beer samples had sufficient residual volume to allow for the testing of a withheld sample. Bubbles occurring when the cell pellets were subjected to hydrogen peroxide would indicate a positive catalase reaction.

**Catalase activity of packaged yeasts.** Catalase tests were performed to examine the physiological states of the oxygen protection pathways of the package-stored yeast and the lab-cultivated yeast (Fig. 4). 19 broadly different ales and lagers were tested immediately after opening and after 3 to 4 days of cultivation (Table 1). Throughout these assays, experimental manipulations were not found to introduce contaminants. Catalase tests were negative, and no colonies were observed from plates inoculated with concentrated sterile beer. Each beer mentioned in this assay, as well as sterile beer spiked with *S. cerevisiae* yeast (WLP001), was found to produce yeast colonies. Bacterial cells were not cultured from the 19 packaged beer samples reported.

All immediate catalase tests were negative. 19 beer styles were tested as triplicate aliquots from 2 to 3 properly packaged units (Table 1; Fig. 5). Cell pellets of these recently opened and concentrated beers did not produce any bubbles when exposed to hydrogen peroxide. This is indicative of deficient catalase neutralization of hydrogen peroxide by the yeast pellets and suggests a suboptimal level of protection from the ROS hydrogen peroxide.

The cell pellets of these 19 beers were successfully aerobically cultured on BTJ agar and were microscopically determined to be viable yeast cells. All cultured catalase assays using these aerobically cultivated colonies were catalase positive (Table 1; Fig. 5). The presence of bubbles denotes the enzymatic ability to neutralize the ROS hydrogen peroxide and confirms the genetic potential of these yeast strains to express catalase.

After aliquoting to perform the immediate and cultured catalase tests in triplicate, if a residual volume of at least 80 mL remained in the whirl bag, then the remaining volume was stored aerobically at room temperature with no additional nutrients for 4 days, at which point its pellet was tested for catalase activity. All withheld catalase assays, the pellets of 6 beers, were catalase positive (Table 1; Fig. 5). The presence of bubbles denotes the enzymatic ability to neutralize the ROS hydrogen peroxide and suggests against the BTJ medium triggering hydrogen peroxide neutralization.

During the procurement of the packaged beers for this study, 2 beer cans were excluded from the standard method described in Fig. 4 due to observed flaws in packaging. These beers exhibited seal issues of their cans, including beer leaking from beneath the seal and substandard can weight (<350 g). The aliquots of these beers were processed and tested for immediate catalase reaction as those of the properly packaged beers were. The pellets from the improperly sealed beers were found to be catalase positive, indicative of a current biochemical ability to neutralize hydrogen peroxide within the "low-fill" cans (Table 1; Fig. 5).

**TABLE 1** Catalase activity results of various packaged beers[a]

| Beer brand | Beer style | Immediate catalase reaction | Cultured catalase reaction | Withheld catalase reaction |
|---|---|---|---|---|
| Bamberger | Mahr's Bräu Unfiltered Lager | Negative | Positive | NT |
| Bell's Brewery Inc. | Official Hazy India Pale Ale (IPA) | Negative | Positive | NT |
| Boulder Beer Company | Hazed and Infused (Pale Ale) | Negative | Positive | NT |
| Boulevard Brewing Company | Unfiltered Wheat | Negative | Positive | Positive |
| Bristol Brewing Company | Beehive: Honey Wheat Ale | Negative | Positive | NT |
| Coors Brewing Company | Coors Light (Lager; Purchased in Golden CO) | Negative | Positive | NT |
| Crazy Mountain Brewery | Hookiebobb IPA | Negative | Positive | NT |
| Denver Beer Company | Love This City (Pilsner) | Negative | Positive | NT |
| Denver Beer Company | Princess Yum Yum (Kolsch) | Negative | Positive | NT |
| Deschutes Brewery | Twilight Summer Ale | Negative | Positive | NT |
| Diebolt Brewing | Anton Francois (French Ale) | Negative | Positive | NT |
| Diebolt Brewing | Postcard Porter | Negative | Positive | NT |
| Good River Beer | Hey, Fishy Fishy (Hazy IPA) | Negative | Positive | Positive |
| Made Here Beer | American Lager | Negative | Positive | NT |
| Prost Brewing Company | Pilsner | Negative | Positive | Positive |
| Ratio Beerworks | Sparks Fly (IPA) | Negative | Positive | Positive |
| Saint Patrick's Brewing Company | Hazy IPA | Negative | Positive | Positive |
| Station 26 Brewing Company | Salt and Lime Mexican Lager | Negative | Positive | Positive |
| Tivoli Brewing Company | Helles Lager | Negative | Positive | NT |
| Positive controls: *S. cerevisiae* yeast (WLP001) from aerobic stock culture | | Positive | Positive | Positive |
| Negative controls: filter-sterilized beer | | Negative | Negative | Negative |
| Undisclosed beer with seal issue | | Positive | NT | NT |
| Undisclosed beer with seal issue | | Positive | NT | NT |

[a]A positive result denotes the presence of bubbles when the cell mass was exposed to hydrogen peroxide. Immediate reaction refers to a centrifuged pellet shortly after package opening, cultured reaction refers to a colony cultivated for 4 days aerobically, and withheld reaction refers to a centrifuged pellet 4 days after package opening. All of the cultured colonies reported were confirmed to be yeast via microscopy. All of the culturing as well as the immediate and cultured catalase results are those of triplicates. All of the withheld catalase results are those of at least duplicates. "NT" denotes that the assay was not tested due to a limited packaged volume or a risk contamination.

## DISCUSSION

Here, we report that different oxygen handling methodologies have significant impacts on viable yeast recovery rates from Bamberger's Mahr's Bräu Unfiltered Lager (BMB). The differences in calculated CFU/mL recovered from the BMB beer bottles ranged from 0.3 ($\pm$0.1) to 533.3 ($\pm$53.6), depending on the oxygen exposure methods employed (Fig. 1). This difference in cultivation success dramatically impacted the volume of sample required to achieve appreciable colonies by over 400-fold. The apparently suboptimal aerobic opening of packaged beer is seemingly sufficient for a large-volume sample that has a high number of viable cells, such as BMB, but may fail with more difficult samples. Our data show that anaerobic opening significantly increases the likelihood of successful yeast recovery from a defined volume of beer.

Facultative organisms, such as *S. cerevisiae*, must operate a perpetual balancing act of utilizing available oxygen while protecting themselves from the threat of ROS damage. Anaerobic opening and microaerophilic incubation seem understandable when thinking of with what a cultivation procedure is requiring *S. cerevisiae* to cope. This condition set

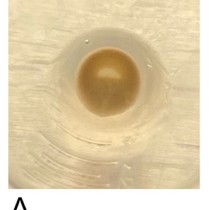 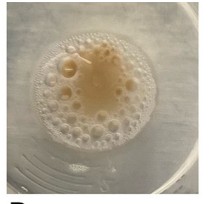 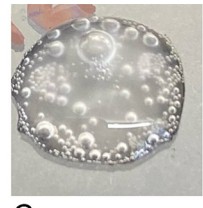 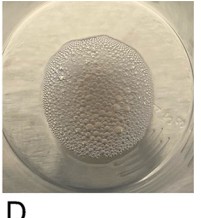

A　　　　B　　　　C　　　　D

**FIG 5** Representative images of the catalase test results. Positive results exhibit bubbles, while negative results do not. (A) The negative result of a properly sealed beer that was immediately pelleted. (B) The positive result of an improperly sealed beer that was immediately pelleted. (C) The positive result of a colony cultured aerobically from a properly sealed beer. (D) The positive result of a properly sealed beer withheld for 4 days and then pelleted.

exhibited an improved cellular recovery rate, which we have found to be associated with anaerobic handling, as well as larger colony sizes, which we have found to be associated with oxygen presence during incubation. Anaerobic opening may protect the cells from an instant influx of ROS, while microaerophilic incubation further protects to some degree as well as supplies some oxygen to promote the healthy growth and persistence of survivors. Cultured subsisters from microaerophilic and anaerobic conditions were able to be transferred to further increased oxygen conditions, and the already cultured cells grew to larger colony sizes with the increased oxygen (Fig. S1D versus Fig. S1E). This environmentally triggered increase in colony size suggests that the small colony size of our anaerobically incubated yeast is an epigenetic, phenotypic variation related to current anaerobic metabolism rather than a stable genetic mutation that prevents oxygen utilization.

Oxygen concentration is known to cascade the regulation of gene expression of more than 25 genes, including oxygen protection genes (15). It has been stated that despite the use of the general term, there is "no such thing as a single general oxidative stress" because ROS impacts each cell in a particular way depending on the condition of that cell (13, 19). ROS, as a whole, lead to impaired physiological function through the cellular damage of DNA, proteins, lipids, and other macromolecules (12). We used DCF dye to show that ROS are present within the yeast of aerobically opened beer, catalase tests to show that those cells currently have suboptimal protection from at least one ROS, and spot-tests to show that ROS negatively impact the growth of bottle-stored yeast. Further biochemical and/or molecular studies could elucidate the causative parameters and/or mechanisms that would complement this culture-based study. We advise that cultivation success is shifted toward higher CFU/mL when bottled beer is opened anaerobically.

Previous bottled-beer cultivation attempts that opened their bottles aerobically may have underreported the viable cell densities within packaged beer due to a bias of aerobic opening somewhat reducing cellular recovery rates. Although this quantitative-culturing data set is from only one beer style, BMB, the catalase data set of 19 beers suggests, at the least, that suboptimal catalase activity may be widespread. Opening beers anaerobically can ensure more confidence in uniquely stringent quality assurance and quality control (QA/QC) screenings of packaged products. The Guidelines of the American Society of Brewing Chemists (20) and other common resources outline widely used atmospheric handling procedures that are reflective of successful common practices of yeast harvesting in breweries and are based on reasonability and efficiency, not necessarily the optimization of cellular recovery. Anaerobic incubations of certain media to detect certain microbes are mentioned, but only the incubation step is directed to be anaerobic in published procedures. Typical QA/QC procedures commonly interpret plates with no growth from aerobically opened and inoculated samples as a green-light for distribution and shelf-stability, as it is deduced that viable cells are not present. Increasing the volumes of beer utilized will always present a more scrutinous evaluation of a sample, which may be necessary with aerobic opening.

Also pertinent, there are rare occasions when ambitious goals of yeast cultivation are being applied to precious fermented samples with long anaerobic incubation times within their vessels, and these may present with small volumes with which scientists must work and/or possibly low viable cell densities (21–25). This literary group includes unique samples of sealed fermented liquid from approximately 1000 BC, 1840 AD, 1890 AD, and 1917 AD recovered from a tomb, shipwreck, museum, and reconstructed brewery, respectively. A microbiologist's holy grail of endeavors, such as these, could be to isolate a viable yeast culture from these beer time capsules and increase the collective yeast library with fascinating strains. Reported methodologies included a sterilized drill bit and a sterile syringe to draw up liquid as the volume is replaced with "sterile air" (25), and others reported the vessels as being "opened" (21, 22) or "extracted" (23) with no explicit mention of anaerobic handling. For future uniquely challenging samples, attempting anaerobic opening may increase the likelihood of successful cultivation. Of course, anaerobic opening cannot resurrect cells that are no longer viable, nor can it ameliorate any lethal conditions, but it could increase the odds of successfully cultivating any viable cell(s) present.

The vast microbial diversity and bioprospecting potential of the rare biosphere and the uncultured majority make the broad pursuit of cultivation optimization paramount to the microbial sciences. Since the dawn of microbiology as a discipline, scientists have recognized the enormous discrepancy between observed microbial richness/diversity and the number of culturable microbes. The various technological advances of the microbial sciences have brought more organisms into culture but have also further elucidated that <1% of microbes are currently estimated to be in culture (26–28). There are many proposed reasons for this, including the general nature of microbes, media, and researchers (26). There is hope that cultivating microbiologists will profoundly consider the epigenetic/biochemical states of the microbes they wish to domesticate, including their nutrient limitations, communication/social cues, and relevant environmental conditions. Here, we present how the environmental condition of oxygen exposure can seemingly impact the recovery and cultivation of a microbial sample. Although one might not consider the rare beer biosphere to be the greatest tapping of the rare biosphere, it would be grand to increase the library of unique microbes involved in brewing. With motivations of not only scientific pursuits but also an intrigued consumer market (2, 6), several topical fermentation scientists' works prompt a general ambition of expanding the repertoire of microbes utilized in pioneering brews (2, 29–39). As new, various, precious samples continue to become assessable, we hope that the prudent means of cultivation presented here aids in the recovery of novel microbial strains.

## MATERIALS AND METHODS

**Yeast recovery from Bamberger's Mahr's Bräu Unfiltered Lager (BMB) with various oxygen exposures.** BMB bottles were used as an unpasteurized, packaged source of beer yeast for quantitative culturing and for comparing different oxygen handling methods. All of the BMB beers for this section's experiments were from the same batch and were procured at the same location. Filter sterilized beer (0.2 $\mu$m) was utilized as a negative-control alongside all aliquoting, concentrating, and culturing of BMB samples to assess proper aseptic technique.

Bottles of BMB were aseptically opened aerobically within a biosafety cabinet or anaerobically within a Bactron300 glove box. The anaerobic state of the glove box was consistently confirmed with resazurin color indicators (Thermo Scientific BR0055B). Each bottle of beer, whether aerobically or anaerobically opened, was poured into a sterile 1 L beaker. This served to help mimic general procedures (20), homogenize the sample to prevent yeast localization, mitigate foaming issues during allocation, and gently "disturb/mix" the beer within that oxygen state. Yeast cells within the beer aliquots were concentrated by centrifugation. Various volumes of beer were used to provide countable culture plates (30 to 300 colonies) (Fig. 1). All volumes of beer were aliquoted into 50 mL sterile centrifuge tube(s) and were centrifuged at 3,234 rcf for 30 min at 4°C. Supernatants greater than 3 mL were discarded from each tube, and cell pellets were resuspended in the remaining ≤3 mL, which was transferred to a 15 mL centrifuge tube. If applicable, this step combined multiple cell slurries from 50 mL tubes into one 15 mL centrifuge tube. These samples were centrifuged at 3,234 rcf for 30 min at 4°C. Most of the supernatant was discarded, and the cell pellet was resuspended in ~100 $\mu$L of the liquid. This entire volume was used as an inoculum for one plate. The anaerobic or aerobic state of each sample was maintained throughout the protocol, including sample aliquoting, decanting, resuspension, and inoculation.

The concentrated yeast was plated on 100 mm plates of BTJ agar (40). Although any culture-based experiment is vulnerable to media bias, BTJ was chosen in an attempt to ameliorate nutrient shock. BTJ medium is presented as the 'General Culture Medium' by the American Society of Brewing Chemists (40). The nutrient state of this medium is a decent proxy for beer, as opposed to many other media that are proxies for wort (e.g., malt extract agar) or are designed to select for beer contaminants rather than for brewer's yeast (e.g., lysine yeast carbon base).

Subsamples of the beer yeast inoculated plates were incubated in different ways: under the originally inoculated oxygen state (aerobic or anaerobic within a Bactron300 glove box) or microaerophilically within a candle jar. The flame within a sealed candle jar reduces the ~18% atmospheric oxygen (at Denver, CO, USA) to 4 to 8% oxygen (at Denver, CO, USA). Due to the presumed slight variation between jars, all directly compared plates were coincubated within the same vessel on the same day to ensure uniform oxygen exposure. With the two different opening/inoculation conditions and the three different incubation conditions, four experimental procedures were evaluated: aerobic/aerobic, aerobic/microaerophilic, anaerobic/microaerophilic, and anaerobic/anaerobic (opened/incubated, respectively).

Triplicate plates were performed for every volume of beer used, and multiple bottles from the same batch were opened as replicates, with a minimum of two bottles. Plates were incubated for 4 days at room temperature, and pictures were taken. For uniformity, ImageJ software (41) was utilized to assess images of these plates in terms of colony counts and sizes after 4 days of incubation.

To test the cultivated yeasts' further tolerance and potential utilization of oxygen, plates with colonies were transferred to increased oxygen conditions. Microaerophilic cultured plates were transferred to

aerobic, and anaerobic cultured plates were transferred to microaerophilic and later to aerobic. Plates were incubated for 4 days at room temperature at each new oxygen condition and were imaged.

**Relevant ecological conditions within bottles of Bamberger's Mahr's Bräu Unfiltered Lager (BMB): microbe(s), oxygen, and reactive oxygen species present.** To enrich and identify microbes within unpasteurized BMB beer, 2 bottles of the same batch were aseptically opened within a biosafety cabinet, decanted into sterile whirl bags, aliquoted, and concentrated via serial centrifugation at 3,234 rcf for 30 min at 4℃. The beer sample pellets were used as a cell mass for a primary enrichment with BTJ broth (40), which was grown up for 4 days at room temperature with gas venting and gentle agitation. These enrichments were used as inocula onto BTJ agar, which were incubated for 4 days at room temperature and assessed for colony and cellular morphology. The BTJ broth enrichments were also pooled as a cell mass for DNA extraction via the Qiagen DNeasy PowerSoil methodology. PCR was performed on the bulk DNA extract using the ITS1 and ITS4 primer set that is commonly used for fungal taxonomy (42), the NL1 and NL4 primer set that is commonly used for ascomycetous yeast taxonomy (43), and the 27F and 1492R primer set that is commonly used for bacterial taxonomy (44). Negative-controls and positive-controls for each primer set were likewise subjected to PCR. The negative-controls were PCR grade water, and the positive controls were a kombucha sample that was previously found to contain yeast and bacteria. Each primer set's potential PCR amplification was performed in triplicate within a Bio-Rad thermocycler with a common protocol of initial denaturation at 98℃ for 5 min, 30 cycles of denaturation at 98℃ for 30 sec, annealing at 55℃ for 30 sec, and extension at 72℃ for 1 min, ending with a final extension at 72℃ for 10 min and visualization using 1% agarose gel electrophoresis. If amplicons were detected via gel electrophoresis, the amplicons were sequenced via Sanger methodology, and each consensus sequence was identified using BLASTn (45). Sequence alignments were performed using ClustalW (46), and neighbor-joining phylogenetic relationships were compiled using MEGA software (47).

An Anton Paar Packaged Beer Analyzer (PBA, Beer Generation M, DMA4500MEC) with a carbo QC ME piercing unit was used to measure the oxygen concentration in three BMB bottled beers (limit of detection ppb). The apparatus also simultaneously measured alcohol, specific gravity, calories, carbon dioxide, and present gravity.

To detect a potential environmental shift from the act of opening the bottle, reactive oxygen species (ROS) within BMB yeast cells was measured. This was accomplished with the epifluorescent dye DCF, as done previously (18). Bottles of BMB beer were aseptically opened, dispensed, and aliquoted either aerobically or anaerobically as described above. With each condition, duplicate bottles were tested, and triplicate aliquots from each bottle were sampled. 5 mL aliquots of BMB beer were centrifuged at 3,234 rcf for 30 min at 4℃ and resuspended in 1 mL of beer supernatant before being supplemented with 10 $\mu$g of DCF. These yeast and DCF mixtures were incubated at 30℃ for 2 h, and the cells were then washed with 1 mL sterile, neutral, phosphate-buffered saline (PBS) and resuspended in 100 $\mu$L of PBS. Anaerobic and aerobic states were maintained during preparation, and then the cover-slipped slides were immediately microscopically observed, unavoidably in regular atmosphere. Images were observed with the FITC HYQ filter of a Keyence BZ-X800 microscope with uniform exposure settings. The general landscape was observed under 400 times magnification, and individual cells were observed under 1,000 times magnification. For uniform quantitative comparisons, ImageJ software was utilized to assess the images of yeast cells and calculate the mean gray values of cells within the green channel (41).

**Impact of reactive oxygen species on Bamberger's Mahr's Bräu Unfiltered Lager (BMB) yeast growth.** To demonstrate bottled yeasts' physiologically relevant sensitivity to ROS, spot-tests were performed, as done previously (48, 49). In general, spot-tests compare the growth capabilities of organisms grown on different agar media by depositing a uniform volume of inoculum and observing the diameter and opacity (integrated density) of the growth spot at the single location of inoculation. YEPD agar (10 g/L yeast extract, 20 g/L peptone, 20 g/L dextrose; pH 6.5) was supplemented with concentrations ranging from 0 to 500 mM. The chemical ROS evaluated were hydrogen peroxide ($H_2O_2$), tert-butyl hydroperoxide (tBHP), menadione, and diamide. Bottles of BMB beer were opened and dispensed anaerobically, as described above. Note that only anaerobically opened beer was used for this assay, so as to approximate a "shock" with ROS in a controlled manner. In triplicate, 15 mL aliquots of anaerobically handled BMB beer were centrifuged at 3,234 rcf for 30 min at 4℃ and resuspended in 1.5 mL of beer supernatant ($\sim$10$\times$ cell concentrate). 4, 8, and 12 $\mu$L of this concentrated slurry was deposited, with triplicate spots, onto standard YEPD plates and on plates with an ROS to compare growth patterns. Plates were incubated anaerobically for 7 days at room temperature and observed. The ImageJ software's dot blot analysis protocol was utilized to uniformly measure the diameters and integrated densities of growth spots when possible (41).

**Catalase activity of bottled yeasts.** The broad assemblage of 19 different packaged beers used for this catalase study was curated to include numerous ale and lager varieties and to exclude styles associated with mixed cultures (e.g., Lambic or Berliner Weiss) and genetic yeast variants (e.g., Saison or Hefeweizen) (Table 1). These disclosed packaged beers were confirmed by weight and physical observation to be adequately filled and properly sealed to industry standards. Only packaged beers shown to have viable/culturable yeast without bacterial contamination were included in this assessment. If cultivation on BTJ agar did not exhibit yeast growth, or if bacterial growth was detected, the data from those undisclosed beers were excluded.

A positive catalase reaction is suggestive of the current enzymatic capability of the cell to neutralize the ROS $H_2O_2$ to mitigate cellular damage. The presence or absence of bubbles when subjected to $H_2O_2$, respectively, denotes a catalase positive or negative reaction for that yeast cell mass at that moment in time. Catalase tests were performed on beer yeast cells at multiple points (Fig. 1): shortly after opening (immediate), after aerobic cultivation with BTJ agar (cultured) (40), and, if possible, after aerobic incubation in unenriched beer for 4 days (withheld).

A wide array of packaged beer with viable yeast was analyzed in this catalase activity survey (Table 1). For each beer type tested, 2 to 3 packaged units from the same batch were sampled, and triplicate aliquots from

each packaged unit were assessed immediately and after culturing. Each packaged beer was aseptically opened, aerobically within a biosafety cabinet, and fully decanted into and homogenized within a sterile whirl bag. Six 40 mL aliquots were pipetted into 50 mL sterile centrifuge tubes for concentration. If >80 mL of the beer liquid remained, then it was sealed with biosafety cabinet atmosphere within the whirl bag and stored at room temperature for 4 days. The six centrifuge tubes that contained beer were immediately centrifuged at 3,234 rcf for 30 min at 4°C and decanted to create cell pellets. Three of the decanted pelleted beer samples were used as a cell mass for an immediate catalase test to survey the microbe's physiological state at that time. All beer liquid was carefully pipetted off the pellet, which was then subjected to 3% $H_2O_2$ and monitored for bubble formation. Three of the decanted pelleted beer samples were used as inocula by resuspending the pellet in ~100 $\mu$L of supernatant and spreading the entire volume onto a BTJ agar plate that was incubated for 4 days at room temperature, aerobically. The colonies that grew from these inoculations were evaluated with methylene blue stain microscopy to confirm yeast viability. These yeast colonies were subjected to a cultured catalase test by aseptically smearing the cells onto plastic, depositing 3% $H_2O_2$, and monitoring for bubble formation. The beer withheld in the whirl bag for 4 days was aliquoted into 50 mL centrifuge tubes, with two to three 40 mL aliquots, depending on the available volume. This pellet was then centrifuged, decanted, and subjected to $H_2O_2$ in the same manner as the immediately handled samples. The presence or absence of bubbles in each catalase assay denoted a catalase positive or negative reaction for that particular cell mass at that moment in time.

For method validation and as a positive-control, filter sterilized beer was spiked with an aerobically cultivated broth culture of beer yeast (*Saccharomyces cerevisiae* WLP001 grown in BTJ broth) and processed in the same way. Concentrated cell pellets were subjected to $H_2O_2$ immediately, while other pellets were inoculated on BTJ plates, the colonies of which were subjected to microscopy and $H_2O_2$. Withheld spiked beer was later concentrated and tested similarly. As a negative-control, filter sterilized beer (0.2 $\mu$m) was also similarly processed with some "pellets" being immediately subjected to $H_2O_2$ while other "pellets" were plated on BTJ plates. Two undisclosed packaged beers with apparent seal issues (beer leaking/bubbling at the seam and a can weight of <350 g) were also similarly assayed with an immediate catalase test on their pellets. These unique packaged products did not have replicate beers but were tested with triplicate aliquots.

## SUPPLEMENTAL MATERIAL

Supplemental material is available online only.
**SUPPLEMENTAL FILE 1**, PDF file, 2 MB.

## ACKNOWLEDGMENTS

Internal funding from the Metropolitan State University of Denver (MSU Denver) supported this research in the form of provost equipment grants, dean's grants, undergraduate research grants, and allocated departmental funding. Support was also awarded by MSU Denver's chapter of the Colorado Wyoming Alliance for Minority Participation (CO-WY AMP) by the National Science Foundation (NSF), award number 1619673. The funders had no role in the study design, data collection and interpretation, or decision to submit the work for publication.

We thank MSU Denver and its Biology department as a whole, especially Fordyce Lux III, Sheryl Zajdowicz, Lisa Gotow, and Mark Karlok. All members of the Ver Eecke research lab are greatly appreciated, especially Frankie Tapia, Alec Rippe, and Andrew Strosnider. MSU Denver's Beverage Analytics QA/QC Laboratory and Katie Strain are acknowledged for collaboration, service, and guidance. Sleeping Giant Brewing Company is sincerely thanked for valuable cooperation. We are also grateful to the Keyence Corporation of America for equipment privileges.

All authors contributed to the conception, data curation, funding acquisition, investigation, and methodology. Kira Pai also contributed via validation and supervision. Helene Ver Eecke also contributed to the study via formal analysis, project administration, resources, supervision, validation, visualization, writing the original draft, and reviewing and editing.

The authors have no conflict of interest to declare.

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
