## [Reviewer comments · Microbiology Spectrum]

Microbiology Spectrum

Increased rate of yeast cultivation from packaged beer with environmentally relevant anaerobic handling.

Kira Pai, Ginger Stout, Theresa Zimmer, Clayton Jacobs, and Helene Ver Eecke

Corresponding Author(s): Helene Ver Eecke, Metropolitan State University of Denver

Review Timeline:

Submission Date:	July 13, 2022
Editorial Decision:	September 6, 2022
Revision Received:	September 27, 2022
Accepted:	September 30, 2022

Editor: Jeffrey Gralnick

Reviewer(s): The reviewers have opted to remain anonymous.

Transaction Report:

DOI: <https://doi.org/10.1128/spectrum.02656-22>

September 6, 2022

Dr. Helene C Ver Eecke
Metropolitan State University of Denver
Biology
Campus Box 53
P.O. Box 173362
Denver, CO 80217

Re: Spectrum02656-22 (Increased rate of yeast cultivation from packaged beer with environmentally relevant anaerobic handling.)

Dear Dr. Helene C Ver Eecke:

Thank you for submitting your manuscript to Microbiology Spectrum. We received one high quality review of your manuscript and rather than continue to try and secure a second reviewer, I would like to give you an opportunity to respond and revise the manuscript. When submitting the revised version of your paper, please provide (1) point-by-point responses to the issues raised by the reviewers as file type "Response to Reviewers," not in your cover letter, and (2) a PDF file that indicates the changes from the original submission (by highlighting or underlining the changes) as file type "Marked Up Manuscript - For Review Only". Please use this link to submit your revised manuscript - we strongly recommend that you submit your paper within the next 60 days or reach out to me. Detailed instructions on submitting your revised paper are below.

Link Not Available

Sincerely,

Jeffrey Gralnick

Senior Editor, Microbiology Spectrum

Journals Department
Editor comments:

Please don't forget to add italics where needed in the references.

Reviewer comments:

Please see attached.

Staff Comments:

Preparing Revision Guidelines

Please return the manuscript within 60 days; if you cannot complete the modification within this time period, please contact me. If you do not wish to modify the manuscript and prefer to submit it to another journal, please notify me of your decision immediately so that the manuscript may be formally withdrawn from consideration by Microbiology Spectrum.

Increased rate of yeast cultivation from packaged beer with environmentally relevant anaerobic handling

The manuscript entitled “Increased rate of yeast cultivation from packaged beer with environmentally relevant anaerobic handling” suggests the anaerobic handling of beer bottles, as well as during the incubation of culture plates, to obtain a high number of CFU/mL. The authors suggested that high levels of ROS during aerobic manipulation negatively impact yeast development. The authors’ hypothesis is confirmed by relatively simple yet efficient tests capable of determining that yeast recovery is affected when comparing the volumes of beer needed to obtain 30-300 colonies (an adequate number of colonies to CFU count) between protocols based on anaerobic or aerobic conditions. The manuscript was well written, clear, within the journal scope, and addresses interesting points of application of the technique. The methodology may be applied by several industries (mainly breweries) in QA/QC routine, as well as in the prospection of new isolates and/or recovery of yeasts from “historical” sources. However, figures need attention, requiring important alterations. Some results need to be further explained, and current references (> 2018) should be added to the text.

Introduction

1. Lines 57-58: As for the presented value of the beer trade, I suggest adding updated data;
2. Lines 65-66: Why is the ecosystem present in a beer bottle considered a relatively unstable community? Since it is usually composed of only one microorganism of only one species.

Results

3. Lines 118-119; 128-130: Could the small colonies identified on the anaerobically incubated plates be *petite mutants*? Could this influence the reuse of these yeasts in future productions using these yeasts?
4. Line 123: How much is “0,3 CFU/mL”? I suggest also presenting the standard deviation values;

5. Line 167: If possible, please add the CFU/mL concentration of the sample from which the “8 μ L” was taken;

Materials and Methods

6. Line 301: How were the bottles stored before they were obtained? They come from unpasteurized beers, correct? Please clarify it in material and methods section;
7. Lines 338-339: Were all the bottles from the same batch?
8. Line 413: It would be very interesting an experiment like this one with a Lambic-style beer, comparing techniques to isolate microorganisms with aerobic and anaerobic incubation (it’s just a suggestion for new experiments)

References

9. Only 3 references are from the last 5 years (total number of references: 48). Therefore, I suggest looking for updated references to highlight the importance and relevance of the study;
10. Scientific names must be in italics (e.g. *Saccharomyces cerevisiae* in some references);

Figures

11. Figure 1: (A) I suggest a reformulation in this figure. Since Y axis has a considerably distance between CFU/mL numbers, it seems “Oxy/Oxy” and “Oxy/Micro” generated no yeast colonies even using ~80-130 mL of beer. The same was observed for “Ana/Micro” and “Ana/Ana” in (C), for which is impossible to determine how many mLs of beer were needed to 30-300 colonies. Separating the Y axis into segments can help.
12. Figure 2A: Please add units on Y axis. Adjust lettering of Y and X axis; I suggest a smaller font size.
13. Lines 170; 175-176; and Figure 3: In this figure, results for tBHP, diamide and manadione were not presented, why?

Increased rate of yeast cultivation from packaged beer with environmentally relevant anaerobic handling

The manuscript entitled “Increased rate of yeast cultivation from packaged beer with environmentally relevant anaerobic handling” suggests the anaerobic handling of beer bottles, as well as during the incubation of culture plates, to obtain a high number of CFU/mL. The authors suggested that high levels of ROS during aerobic manipulation negatively impact yeast development. The authors’ hypothesis is confirmed by relatively simple yet efficient tests capable of determining that yeast recovery is affected when comparing the volumes of beer needed to obtain 30-300 colonies (an adequate number of colonies to CFU count) between protocols based on anaerobic or aerobic conditions. The manuscript was well written, clear, within the journal scope, and addresses interesting points of application of the technique. The methodology may be applied by several industries (mainly breweries) in QA/QC routine, as well as in the prospection of new isolates and/or recovery of yeasts from “historical” sources. However, figures need attention, requiring important alterations. Some results need to be further explained, and current references (> 2018) should be added to the text.

We thank the reviewer for their time, contemplative evaluation, concise summary, and valuable advice. Beneficial updates to the manuscript have been completed based on the reviewer’s comments:

Introduction

1. Lines 57-58: As for the presented value of the beer trade, I suggest adding updated data;

2021 data has been derived directly from the United Nations Comtrade Database and an up-to-date estimate has been included (line 58).

2. Lines 65-66: Why is the ecosystem present in a beer bottle considered a relatively unstable community? Since it is usually composed of only one microorganism of only one species.

The original verbiage oversimplified the ecological theory inversely relating species richness with ecosystem stability based on reduced functional redundancy. The text has been modified to be more elaborative and hopefully more accessible to more readers (line 64).

“Ecological theory inversely relates the number of species and their functional redundancies to ecological stability (8). A monocultured/single-species beer would probably be significantly impacted by environmental shifts due to being unlikely to be resilient on account of minimal functional redundancy.”

Results

3. Lines 118-119; 128-130: Could the small colonies identified on the anaerobically incubated plates be petite mutants? Could this influence the reuse of these yeasts in future productions using these yeasts?

Petite mutants are a genetic variant that yield small colony size essentially due to the genetic inability to grow aerobically even when oxygen is present (they can only grow anaerobically due to loss of function of the mitochondrial and/or host genes for oxidative phosphorylation). We forced yeasts to grow anaerobically thus shifting them into a similar state as petit mutants. But our small colony yeasts are not stable mutants that have lost the function to use oxygen, they've been deprived of oxygen. This reviewer's comment is a very valuable cue to elaborate more about epigenetic shifts versus genetic mutations. The small colony sizes of our anaerobic incubated yeasts are an epigenetic phenotype, not related to a genetic mutation. This is showcased to be epigenetic when the colony size increases when a previously anaerobically incubated plate is later exposed to oxygen. If the yeasts were genetic mutants unable to oxidatively phosphorylate, then they would stay their small size even when increased oxygen is introduced. Elaborative comments have been added to the manuscript (lines 237-239, & 350) to orient the appreciation of epigenetics versus mutations.

"This environmentally triggered increase in colony size suggests that the small colony size of our anaerobically incubated yeast is an epigenetic phenotypic variation related to current anaerobic metabolism, rather than a stable genetic mutation preventing oxygen utilization."

"To test the cultivated yeasts' further tolerance and potential utilization of oxygen, plates with colonies were transferred to increased oxygen conditions."

4. Line 123: How much is "0,3 CFU/mL"? I suggest also presenting the standard deviation values;

0.3 CFU/ml, relates to the mathematical calculation that 80-120 ml of beer produced 30-300 colonies. One ml of beer would not produce even one colony, on average, so beer volumes tested needed to be increased and mathematically manipulated to the standard unit of per ml. The text has been edited to be more elaborative.

"Countable plates were achieved for all conditions, average colony counts were quantified, and average CFU/ml were calculated based on the beer volume utilized" (line 119)

Standard deviation values are warranted and have been added to the text (lines 122-126, 217-219).

5. Line 167: If possible, please add the CFU/mL concentration of the sample from which the "8 μ L" was taken;

A CFU/ml was not enumerated for the spot test technique because the sample was not spread across the 100m plate, as one would for a traditional spread plate for CFU enumeration. For this assay, inocula are deposited on a spot, rather than the whole plate. Standard dot-blot analysis of spot tests was performed for each growth spot, and data enumerated with the standard integrated density rather than CFU/ml. Each spot's integrated density was related to that of the control (YEPD standard medium).

We tried to explain the general procedure of spot-tests that are commonly done:

"In general, spot-tests compare the growth capabilities of organisms grown on different agar media by depositing a uniform volume of inoculum and observing the diameter and

opacity (integrated density) of the growth spot at the single location of inoculation” (line 400)

“Conventional spot-tests were performed by depositing a uniform volume of inoculum and observing the diameter and opacity (integrated density) of the growth spot on agar after incubation.” (line 167)

Materials and Methods

6. Line 301: How were the bottles stored before they were obtained? They come from unpasteurized beers, correct? Please clarify it in material and methods section;

Yes, beers were unpasteurized and unfiltered, as a source of viable yeast. This has been modified to be more elaborative (line 374-309, 358).

7. Lines 338-339: Were all the bottles from the same batch?

Yes, for each methodological section all bottles were from the batch obtained from the same location, attempting to have comparative samples being compared by the same methodology. This has been modified to be more elaborative (lines 309, 343, 346, 359, & 437)

8. Line 413: It would be very interesting an experiment like this one with a Lambic-style beer, comparing techniques to isolate microorganisms with aerobic and anaerobic incubation (it’s just a suggestion for new experiments)

Indeed, mixed fermentation samples of Lambic and/or hard-kombucha are fascinating future directions.

References

9. Only 3 references are from the last 5 years (total number of references: 48). Therefore, I suggest looking for updated references to highlight the importance and relevance of the study;

Thank you for this insight. This is a valuable insight on a manuscript a long time in the making and a relatively infrequently published microbial subdiscipline. I have added several references to highlight current knowledge and applications such as:

- *United Nations Commodity Trade Statistics Database (2021). <https://comtrade.un.org/>. Retrieved 12 September 2022.*
- *Villacreces S, Blanco CA, Caballero I. 2022. Developments and characteristics of craft beer production processes. Food Bioscience 45:101495*
- *Barnette BM, Shellhammer TH. 2019. Evaluating the Impact of Dissolved Oxygen and Aging on Dry-Hopped Aroma Stability in Beer. Journal of the American Society of Brewing Chemists 77:179–187.*
- *Sachdev S, Ansari SA, Ansari MI, Fujita M, Hasanuzzaman M. 2021. Abiotic Stress and Reactive Oxygen Species: Generation, Signaling, and Defense Mechanisms. 2. Antioxidants 10:277.*
- *Lewis WH, Tahon G, Geesink P, Sousa DZ, Ettema TJG. 2021. Innovations to culturing the uncultured microbial majority. 4. Nat Rev Microbiol 19:225–240.*

- *Einfalt D. 2021. Barley-sorghum craft beer production with Saccharomyces cerevisiae, Torulaspora delbrueckii and Metschnikowia pulcherrima yeast strains. Eur Food Res Technol 247:385–393.*
- *Callejo MJ, González C, Morata A. 2017. Use of Non-Saccharomyces Yeasts in Bottle Fermentation of Aged Beers. Brewing Technology. InTech.*
- *Biggs CR, Yeager LA, Bolser DG, Bonsell C, Dichiera AM, Hou Z, Keyser SR, Khursigara AJ, Lu K, Muth AF, Negrete Jr. B, Erisman BE. 2020. Does functional redundancy affect ecological stability and resilience? A review and meta-analysis. Ecosphere 11:e03184.*

10. Scientific names must be in italics (e.g. *Saccharomyces cerevisiae* in some references);

Indeed. The reference section's text has been modified accordingly.

Figures

11. Figure 1: (A) I suggest a reformulation in this figure. Since Y axis has a considerably distance between CFU/mL numbers, it seems “Oxy/Oxy” and “Oxy/Micro” generated no yeast colonies even using ~80-130 mL of beer. The same was observed for “Ana/Micro” and “Ana/Ana” in (C), for which is impossible to determine how many mLs of beer were needed to 30-300 colonies. Separating the Y axis into segments can help.

The figure and figure caption have been modified. The axes have been ‘broken’ as the author suggests. One can see the data isn’t at the zero mark, but it is still quite a dramatic difference in values. The figure caption now includes numerical data for more precise numerical data reference.

12. Figure 2A: Please add units on Y axis. Adjust lettering of Y and X axis; I suggest a smaller font size.

Modifying the font size has occurred. I don’t believe there is a standardized unit for this type of assay. I’ve altered the axes to read ‘fluorescence intensity,’ which seems to be the most common axes title for published DCF fluorescence.

13. Lines 170; 175-176; and Figure 3: In this figure, results for tBHP, diamide and manadione were not presented, why?

The spot tests looked remarkably similar across various ROS and replicate spots, and hydrogen peroxide was seemingly the most relevant due to its direct relation to the catalase test. This use of representative images was motivated by being concise and was meant to streamline the figure/data. I have modified the figure to an alternative version with representative images of all ROS tested and updated the figure caption for panels A-I. They are still representative images as the replicates performed and analyzed are not all visualized, but each ROS is now visualized.

September 30, 2022

Dr. Helene C Ver Eecke
Metropolitan State University of Denver
Biology
Campus Box 53
P.O. Box 173362
Denver, CO 80217

Re: Spectrum02656-22R1 (Increased rate of yeast cultivation from packaged beer with environmentally relevant anaerobic handling.)

Dear Dr. Helene C Ver Eecke:

Your manuscript has been accepted, and I am forwarding it to the ASM Journals Department for publication. You will be notified when your proofs are ready to be viewed. (As a beer nerd, I really enjoyed reading your paper!!)

Sincerely,

Jeffrey Gralnick
Editor, Microbiology Spectrum
